# The Devil Is in the Details: Tackling Unimodal Spurious Correlations for Generalizable Multimodal Reward Models

Zichao Li [1 2]   Xueru Wen [1 2]   Jie Lou [3]   Yuqiu Ji [3]   Yaojie Lu [1]   Xianpei Han [1]   Debing Zhang [3]   Le Sun [1]

## Abstract

Multimodal Reward Models (MM-RMs) are crucial for aligning Large Language Models (LLMs) with human preferences, particularly as LLMs increasingly interact with multimodal data. However, we find that MM-RMs trained on existing datasets often struggle to generalize to out-of-distribution data due to their reliance on unimodal spurious correlations, primarily text-only shortcuts within the training distribution, which prevents them from leveraging true multimodal reward functions. To address this, we introduce a Shortcut-aware MM-RM learning algorithm that mitigates this issue by dynamically reweighting training samples, shifting the distribution toward better multimodal understanding, and reducing dependence on unimodal spurious correlations. Our experiments demonstrate significant improvements in generalization, downstream task performance, and scalability, establishing a more robust framework for multimodal reward modeling. Our source code is provided on https://github.com/alignrm/Generalizable-MM-RM.

## 1. Introduction

Reward Models (RMs) (Stiennon et al., 2020; Ouyang et al., 2022), serving as proxies for human preferences, are crucial in Large Language Models (LLMs) alignment (Achiam et al., 2023; Touvron et al., 2023). As LLMs increasingly perceive the world in multimodal ways (Yin et al., 2023), multimodal reward modeling (Sun et al., 2023; Amirloo et al., 2024; Yan et al., 2025) has become essential for addressing challenges like visual hallucinations (Zhou et al.,

This work was conducted during Zichao Li and Xueru Wen's internship at Xiaohongshu. [1]Chinese Information Processing Laboratory, Institute of Software, Chinese Academy of Sciences [2]University of Chinese Academy of Sciences [3]Xiaohongshu Inc. Correspondence to: Jie Lou <loujie0822@gmail.com>, Yaojie Lu <luyaojie@iscas.ac.cn>.

*Proceedings of the 42nd International Conference on Machine Learning*, Vancouver, Canada. PMLR 267, 2025. Copyright 2025 by the author(s).

2023) and safety concerns (Liu et al., 2025), making it a crucial step toward more reliable and aligned AI systems.

The effectiveness of Multimodal Reward Models (MM-RMs) hinges on their ability to generalize and adapt to diverse inputs, as generalization is critical for ensuring their reliability (Yang et al., 2024; Dou et al., 2024). If an RM fails to generalize to out-of-distribution (*o.o.d.*) during RL training, the policy model may prioritize maximizing rewards at the expense of aligning with human intent (Gao et al., 2023), potentially leading to reward hacking (Amodei et al., 2016; Pan et al., 2022). Furthermore, poor generalization undermines the trustworthiness of an RM on unseen data, hindering test time scaling strategies (Snell et al., 2024; Luo et al., 2024) such as selecting optimal responses from candidate outputs. Therefore, understanding and improving the generalization capabilities of MM-RMs is essential for ensuring their robustness in real-world scenarios.

In this work, we identify a key limitation of Multimodal Reward Models (MM-RMs) trained on existing multimodal preference datasets: their inability to generalize effectively due to unimodal spurious correlations. **These correlations occur when models over-rely on non-robust text-only patterns that hold within the training distribution but fail to generalize to out-of-distribution (*o.o.d.*) scenarios.** To investigate this issue, we conduct cross-distribution transfer experiments across diverse multimodal preference datasets. Our results reveal a significant drop in MM-RMs' accuracy on *o.o.d.* data, while their performance remains stable in *i.i.d.* settings. To further analyze the impact of text-only shortcuts on generalization, we propose the Shortcut-Failure Degradation (SFD) metric, which measures the performance drop of MM-RMs when unimodal correlations fail to generalize. Our results demonstrate a critical flaw: **Even in multimodal training settings, MM-RMs often default to text-only shortcuts, limiting their ability to learn generalizable multimodal reward functions**. This suggests that existing datasets and training strategies may not sufficiently encourage true multimodal understanding in MM-RMs.

To address this challenge, **we propose a Shortcut-aware MM-RM learning algorithm, designed to mitigate the reliance on unimodal spurious correlations in the presence of biased training data.** Building on the discovery of uni-

modal shortcuts, our method introduces a text-only RM as a shortcut-proxy during training. **This proxy explicitly identifies instances where text-only shortcuts fail, enabling dynamic sample reweighting to shift the training distribution toward scenarios where multimodal understanding is essential.** Experiments show that our algorithm significantly improves generalization in cross-distribution transfer evaluations, reducing dependence on unimodal spurious correlations. Furthermore, it achieves stronger performance and scalability in downstream test-time tasks, demonstrating its robustness and practical applicability.

To conclude, our primary contributions are three-fold:

- We identify unimodal spurious correlations as a critical challenge in MM-RM development, demonstrating that they undermine MM-RM's generalization on unseen data, even in multimodal training settings.

- A Shortcut-aware MM-RM learning algorithm is proposed, which mitigates MM-RMs' reliance on unimodal spurious correlations by dynamically reweighting samples to emphasize multimodal understanding.

- Our algorithm achieves generalization improvements across various biased training sets, along with enhanced downstream performance and scalability, setting a more robust paradigm for MM-RM construction.

## 2. Preliminary

### 2.1. Multimodal Reward Modeling

Throughout our paper, we formalize the problem of multimodal reward modeling in a pairwise preference labeling (Ouyang et al., 2022) paradigm. Consider a preference dataset $\mathcal{S} := \{(\boldsymbol{x}_i, y_i)\}_{i=1}^n$, where $\boldsymbol{x}_i$ can be denoted as $\boldsymbol{x}_i = (v_i, q_i, a_i^1, a_i^{-1})$ and $y_i \in \{1, -1\}$. Here, $v_i$ denotes an image, residing in the vision space $\mathbb{V}$, and $q_i$, $a_i^1$, and $a_i^{-1}$ correspond to a query and two distinct answers, all within a shared language space $\mathbb{T}$. We thereby define $t_i = (q_i, a_i^1, a_i^{-1})$. The preference label $y_i$ identifies the chosen answer index given the query $q_i$ and the image $v_i$.

We consider a multimodal reward model $\mathcal{M}$ as a pointwise predictor that maps the provided input (consisting of an image, a query and an answer) to a numerical score, denoted as $r : \mathbb{V} \times \mathbb{L} \to \mathbb{R}$. At the training stage, we follow the standard Bradley-Terry Model (Bradley & Terry, 1952) paradigm, optimizing the MM-RM by minimizing the empirical loss of $r$ over the training set $\mathcal{S}_{train}$ as follows:

$$\mathcal{L} = - \mathop{\mathbb{E}}_{(\boldsymbol{x}_i, y_i) \in \mathcal{S}_{train}} [\log(\sigma(r(v_i, q_i, a_i^{y_i}) - r(v_i, q_i, a_i^{-y_i})))]$$

$$(1)$$

In the evaluation phase, given the test set $\mathcal{S}_{test}$, we adhere to the typical setting (Lambert et al., 2024), using pairwise comparison accuracy as the RM performance metric:

$$\text{Acc} = \mathop{\mathbb{E}}_{(\boldsymbol{x}_i, y_i) \in \mathcal{S}_{test}} \mathbb{I}(r(v_i, q_i, a_i^{y_i}) > r(v_i, q_i, a_i^{-y_i})))$$

$$(2)$$

### 2.2. Cross-Distribution Generalization

In the classical machine learning setting, both the training set $\mathcal{S}_{train}$ and test set $\mathcal{S}_{test}$ are *i.i.d.* samples drawn from the same environment, i.e., there exists a distribution $\mathcal{D} \sim \mathbb{P}(X, Y)$ such that $\mathcal{S}_{train} \overset{i.i.d.}{\sim} \mathcal{D}$ and $\mathcal{S}_{test} \overset{i.i.d.}{\sim} \mathcal{D}$. To systematically explore the generalization behavior of MM-RMs on unseen but related environments (Arjovsky et al., 2019), we now introduce our cross-distribution framework.

Specifically, we consider a collection of distinct distributions $\{\mathcal{D}^e \sim \mathbb{P}^e(X^e, Y^e)\}_{e \in \mathcal{E}}$ obtained from a set of environments $\mathcal{E}$, where $e$ is the the environment index. Despite varying in factors like annotation methods and sampling models, different environments share common objectives: the pursuit of faithful and helpful responses. More formally, each environment satisfies the *Invariance Condition* (Ahuja et al., 2020) assumption, i.e., there exists a representation that is common to all environments, allowing for consistent label prediction across different probability distributions.

We thus obtain distinct datasets $\{\mathcal{S}^e \overset{i.i.d.}{\sim} \mathcal{D}^e\}$, where each $\mathcal{S}^e$ can be divided into $\mathcal{S}_{train}^e$ and $\mathcal{S}_{test}^e$. **This allows us to study the generalization performance of multimodal reward models under both *i.i.d.* and *o.o.d.* scenarios**: We restrict the training data to any single environment, denoted as $\mathcal{S}_{train}^e$, then for model $\mathcal{M}^e$ trained on this dataset, $\mathcal{S}_{test}^e$ serves as the *i.i.d.* test scenario, while any $\mathcal{S}_{test}^{e'}$ where $e' \neq e$ constitutes *o.o.d.* test scenarios. By evaluating $\mathcal{M}^e$'s performance variations across these scenarios, we gain insights into its robustness and generalization.

### 2.3. Unimodal Spurious Correlations

Previous work attributes machines' poor generalization to spurious correlations, also known as shortcut learning (Simon, 1954; Geirhos et al., 2020). While machines excel at learning correlations between inputs and labels, some of these correlations act as shortcuts that hold only within the training distribution, failing to generalize to unseen data.

In this paper, we identify a specific type of spurious correlations in MM-RMs, termed **unimodal spurious correlations**. Following Cadene et al., we use "unimodal" to refer to "text-only". More specifically, unimodal spurious correlations refer to text-only shortcuts learned by MM-RMs on training data, which achieve unexpected success in *i.i.d.* scenarios but fail in *o.o.d.* settings (see Section 3.3), thereby hindering MM-RMs' generalization to unseen data (see Section 3.4).

To systematically analyze and diagnose unimodal spurious

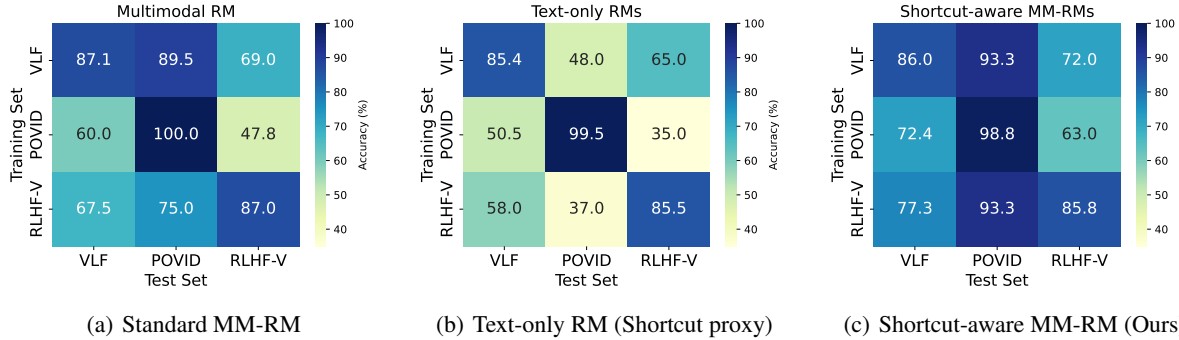

(a) Standard MM-RM        (b) Text-only RM (Shortcut proxy)        (c) Shortcut-aware MM-RM (Ours)

*Figure 1.* Cross-distribution evaluations of three kinds of RM, where the diagonal elements represent *i.i.d.* tests, while the off-diagonal elements represent *o.o.d.* tests. (a) Standard MM-RM has significant room for improvement in certain *o.o.d.* test scenarios. (b) Text-only shortcuts achieve high accuracy under *i.i.d.* tests but demonstrate poor generalization in *o.o.d.* scenarios. (c) **Our algorithm demonstrates substantial improvements in generalization, with average accuracy across six *o.o.d.* scenarios increasing from 68.1 to 78.5.**

correlations, we introduce two specialized text-only settings:

(1) *Text-only Training*, i.e., training the reward models after removing the visual modality from the preference datasets, modifying the loss function in Equation (1) as shown below:

$$\mathcal{L}_t = - \mathop{\mathbb{E}}_{(\boldsymbol{x}_i, y_i) \in \mathcal{S}_{train}} \left[ \log(\sigma(r_t(q_i, a_i^{y_i}) - r_t(q_i, a_i^{-y_i}))) \right] \tag{3}$$

(2) *Text-only Test*, which involves removing the visual modality during the inference phase on the test set, thereby modifying the accuracy metric in Equation (2) as follows:

$$\text{Acc}_t = \mathop{\mathbb{E}}_{(\boldsymbol{x}_i, y_i) \in \mathcal{S}_{test}} \mathbb{I}(r_t(q_i, a_i^{y_i}) > r_t(q_i, a_i^{-y_i})) \tag{4}$$

## 3. Generalization and Unimodal Spurious Correlations of Multimodal Reward Models

### 3.1. Setup

**Data.** We utilize three existing vision-language preference datasets: VLFeedback (Li et al., 2024b), POVID (Zhou et al., 2024), and RLHF-V (Yu et al., 2024), which have been used to construct MM-RMs in previous work (Amirloo et al., 2024). Each dataset is split into training and test sets to represent distribution drift across distinct environments. These datasets focus on vision-related tasks like VQA and image captioning, but differ in construction methods. Additional details of data setup are shown in Appendix A.1.

**Model.** We primarily utilize InternVL2-8B (Chen et al., 2024b) as the basis for reward model construction. To adapt the Vision-Lanugage Model to an MM-RM architecture, we remove the language modeling head $l$ and add an RM head $h$, which is a linear layer that converts the hidden states from the last layer of the LLM decoder into a scalar reward score. Additional details regarding model architecture and training hyperparameters are presented in Appendix A.2.

### 3.2. How Well Does MM-RM Generalize?

We conduct the cross-distribution tests and present the results in the form of a generalization matrix (see Figure 1(a)), i.e., the $(e, e')$ element in the matrix represents the performance of the model trained on $\mathcal{S}_{train}^e$ when tested on $\mathcal{S}_{test}^{e'}$.

**Insight 1: Standard MM-RMs exhibit a marked decline in accuracy on *o.o.d.* data compared to *i.i.d.* scenarios.** We first analyze the average accuracy across three *i.i.d.* test scenarios and six *o.o.d.* test scenarios (91.4 vs 68.1), revealing a non-negligible performance gap of 23.2 (see Figure 1(a)). Moreover, the training set has a significant impact on the robustness of MM-RMs. The MM-RM trained on VLFeedback demonstrates relatively good generalization; however, there are still environments where the model fails to transfer successfully, such as a 18.1 accuracy drop on RLHF-V. POVID represents the least generalizable training data, which perfectly fits *i.i.d.* data (100.0) but results in below-random accuracy (47.8) in certain *o.o.d.* scenarios.

We are interested in understanding the performance degradation of MM-RMs in *o.o.d.* scenarios. Although training data size is a consideration, we demonstrate through scaling experiments on VLFeedback that size is not the decisive factor, as detailed in Appendix B. In this work, we identify unimodal spurious correlations as a more subtle yet critical issue inherent in current MM-RMs' construction (see Section 3.3), which significantly hinders their generalization on unseen data (see Section 3.4).

### 3.3. MM-RMs Learn to Exploit Text-only Shortcuts

We investigate text-only shortcuts as a non-negligible form of spurious correlations in MM-RMs' construction and draw the following insights.

**Insight 2: Existing multimodal preference datasets inevitably exhibit text-only shortcuts, which only hold in

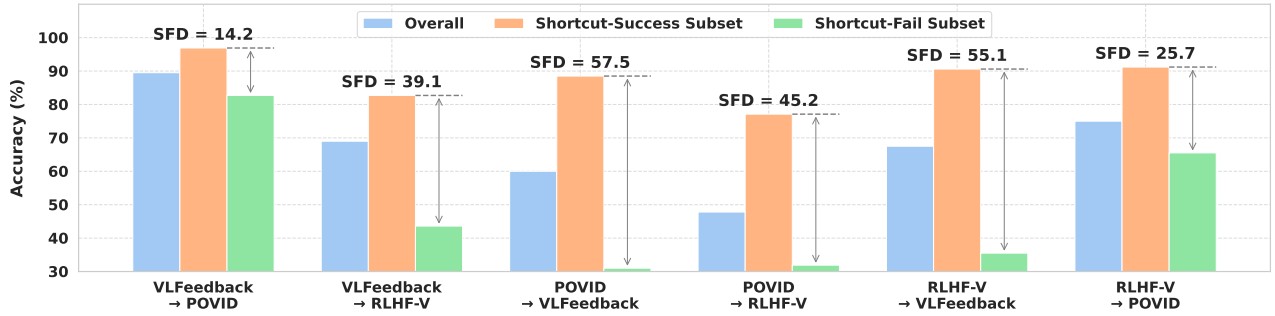

*Figure 2.* Accuracy and Shortcut-Failure Degradation of MM-RMs in various *o.o.d.* scenarios. $\mathcal{S}^e \to \mathcal{S}^{e'}$ indicates that the MM-RM trained on $\mathcal{S}^e_{train}$ is tested on $\mathcal{S}^{e'}_{test}$. **MM-RMs show consistently poor performance when text-only shortcuts become ineffective.**.

*Table 1.* Accuracy of multimodal reward models and its text-only variants on *i.i.d.* scenarios. **Image removal at any stage results in only marginal in-distribution accuracy degradation.**

| Text-only | | Dataset | | | Avg. Drop |
|:---:|:---:|:---:|:---:|:---:|:---:|
| *Train* | *Test* | *VLF* | *POVID* | *RLHF-V* | |
| ✗ | ✗ | 87.1 | 100.0 | 87.0 | - |
| ✓ | ✓ | 85.4 | 99.5 | 85.5 | 1.2 |
| ✓ | ✗ | 86.2 | 99.8 | 84.0 | 1.4 |
| ✗ | ✓ | 82.5 | 98.8 | 84.3 | 2.8 |

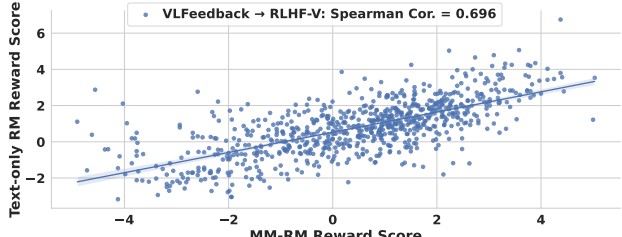

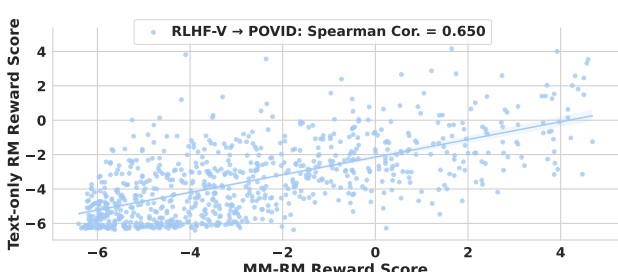

*Figure 3.* The correlation between MM-RM reward scores and text-only RM scores, using two *o.o.d.* test scenarios as examples.

**their corresponding distributions.** Based on text-only training and testing settings, we derive the generalization matrix of text-only RM (see Figure 1(b)). The text-only model achieves comparable accuracy to the standard MM-RM under *i.i.d.* conditions across all data distributions. Conversely, text-only shortcuts fail in *o.o.d.* scenarios, which indicates that these shortcuts merely serve as spurious correlations valid only under specific distributions, rather than the genuine rewarding functions that the model should learn.

**Insight 3: Even when trained on multimodal preference environments, MM-RMs still learn to exploit unimodal spurious correlations.** We alternate between multimodal and text-only modes during training and testing, examining the reward model's performance under *i.i.d.* conditions (see Table 1). Here, we find that models trained on multimodal preference data still achieve comparable in-distribution performance during text-only evaluation, indicating the presence of text-only shortcuts in their learned correlations.

These insights suggest that the poor generalization of MM-RMs may be partially attributed to their acquisition of unimodal spurious correlations during multimodal training.

### 3.4. Shortcuts Hinder MM-RMs' Generalization

To systematically examine the impact of text-only shortcuts on MM-RMs' generalization, we propose the Shortcut-Failure Degradation (SFD) metric, which quantifies the performance drop of MM-RMs when unimodal spurious correlations fail to generalize to *o.o.d.* data.

**Definition 3.1.** (Shortcut-Failure Degradation). Given a standard MM-RM $\mathcal{M}^e$ trained on dataset $\mathcal{S}^e_{train}$, we employ the text-only RM $\mathcal{M}^e_t$ as a proxy for text-only shortcuts, which is initialized with the same model weights and trained on the identical dataset $\mathcal{S}^e_{train}$. This text-only RM is then used to split an *o.o.d.* test set $\mathcal{S}^{e'}_{test}$ into two parts: one part is successfully classified by the text-only RM, termed the *shortcut-success subset*, while the other part, where the text-

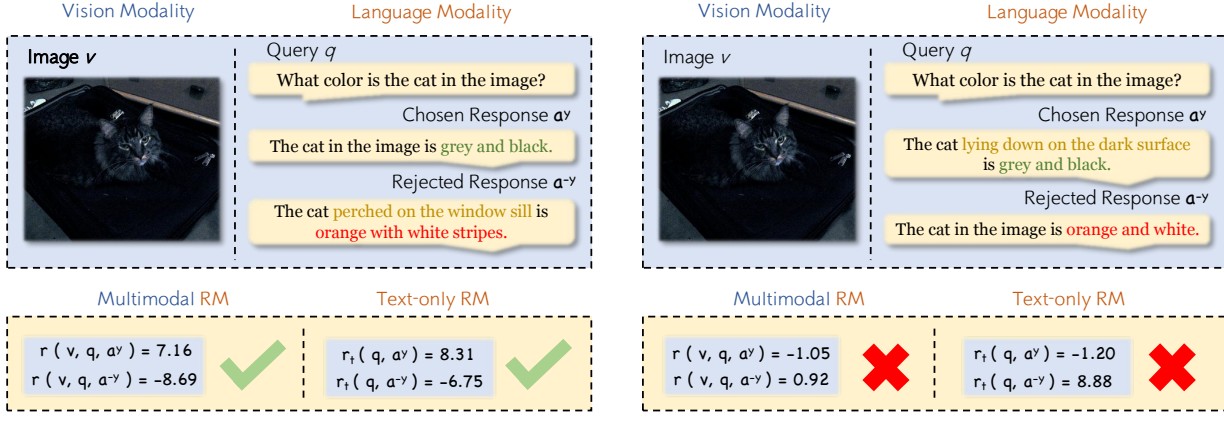

(a) An example of POVID *i.i.d.* tests. The models learn unimodal spurious correlations between query-irrelevant descriptions and bad responses, rather than developing a genuine rewarding of "right versus wrong" that requires multimodal understanding.

(b) A simulated example of *o.o.d.* scenarios where we create a stress test by slightly modifying two responses. Both MM-RM and text-only RM fail to assign higher scores to the chosen response, demonstrating a lack of robustness.

*Figure 4.* An empirically observed text-only shortcuts in POVID data, which generates rejected responses by injecting hallucinations into standard answers, often introducing spurious correlations between query-irrelevant descriptive elements and bad responses.

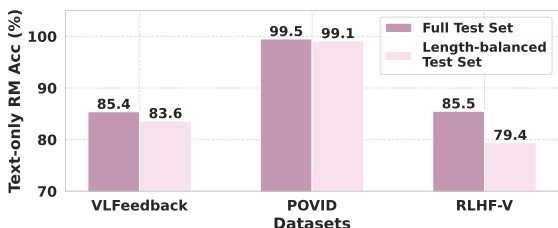

*Figure 5.* Text-only RMs achieve high accuracy on both the full test set and a length-balanced subset under *i.i.d.* scenarios. **Besides length bias, there still remain fine-grained unimodal spurious correlations that can be learned by models.**

only RM fails, is termed the *shortcut-fail subset*. Shortcut-Failure Degradation refers to the difference in accuracy achieved by the standard MM-RM between the *shortcut-success subset* and the *shortcut-fail subset*. [1]

**Insight 4: The generalization capability of MM-RMs is heavily constrained by the unimodal spurious correlations.** The SFD values of MM-RMs range from 14.2 to 57.5 across different out-of-distribution scenarios, with an average of 39.5 (see Figure 2). This reveals that MM-RMs' reward-scoring process is heavily dominated by text-only shortcuts. When these shortcuts fail to generalize to *o.o.d.* data, particularly in scenarios requiring true multimodal understanding, the models exhibit substantial performance degradation. We further validate this observation by analyzing the correlation between the reward scores of MM-RMs and text-only RMs (see Figure 3).

---

[1]We further analyze the implications of SFD in Appendix C.

## 3.5. Where Do The Unimodal Spurious Correlations of MM-RMs Arise From?

Given our statistical findings that MM-RMs tend to learn text-only shortcuts, leading to poor generalization, we seek to deepen our insights of the following question: *Where do the unimodal spurious correlations of MM-RMs arise from?*

### 3.5.1. SHORTCUT OPPORTUNITIES WITHIN DATASETS

Data biases (Bahng et al., 2020; Jung et al., 2023), or high-frequency co-occurrences of irrelevant features with labels, create opportunities for spurious correlations (Agrawal et al., 2018; Cadene et al., 2019). In reward modeling, length bias (Shen et al., 2023; Chen et al., 2024a) is the most intuitive example: In preference datasets, longer and well-structured responses are often preferred, leading to a learnable bias that diverges from human expectations.

We first analyze the length characteristics of each training set, where the proportion of chosen responses being longer is 31.5% (POVID), 59.8% (VLFeedback), and 67.8% (RLHF-V), respectively. Then, we downsample each test set to achieve length parity between chosen and rejected responses, and observe the performance changes of text-only RMs, which serve as a proxy for unimodal shortcuts (see Figure 5). We observe a slight decrease in the accuracy of text-only RMs, but they still maintain unreasonably high performance, indicating that there still remain fine-grained unimodal spurious correlations that can be learned.

We further consider the fine-grained spurious correlations from a case study perspective, using POVID data as an example (see Figure 4). As POVID generates rejected responses by injecting hallucinations into standard answers,

*Figure 6.* An overview of our Shortcut-aware MM-RM learning paradigm.

this process often introduces query-irrelevant descriptive elements. Consequently, the models capture unimodal spurious correlations between query-irrelevant descriptions and bad responses, rather than developing a genuine multimodal understanding. Once there are slight perturbations in the response text, the multimodal reward model fails, reflecting its inability to adapt to dynamic real-world environments.

While one might consider eliminating such correlations in the data annotation process, we argue this is impractical in practice. Even large-scale, diverse datasets inherently contain numerous shortcuts that may be unobservable, even to humans (Calude & Longo, 2017; Geirhos et al., 2020). Instead of explicitly modifying the datasets, our work implements implicit dynamic debiasing of the data, which is applicable to any multimodal preference datasets. The detailed methodology will be presented in Section 4.

### 3.5.2. FEATURE ACQUISITION UNDER LIMITED REPRESENTATION CAPACITY

As datasets create opportunities for unimodal spurious correlations, how these correlations are learned also depends on the model's limited representation capacity (Yang et al., 2022). Under empirical risk minimization, models tend to prioritize learning easily acquired features (Vapnik, 1991; Hermann et al., 2023), such as text-only shortcuts in our context. This can be explained by the modality gap (Liang et al., 2022) in multimodal models: a fixed-size image contains vast fine-grained information (e.g., shapes, textures, and backgrounds), creating a significant information imbalance compared to textual content (Schrodi et al., 2024).

To verify this, we conduct investigation experiments using visual patch numbers as proxies for feature acquisition difficulty, with results presented and discussed in Appendix D. While results confirm their correlation with MM-RMs' generalization, we note that simply scaling the visual patch numbers cannot serve as a universal solution for all data scenarios, and there exists a notable performance bottleneck, highlighting the importance of a systematic algorithm to address unimodal spurious correlations.

## 4. Shortcut-Aware Multimodal RM

Previous sections demonstrate that MM-RMs tend to exploit unimodal spurious correlations during multimodal learning, which hinder their ability to perform generalizable multimodal reward-scoring. A key insight emerges from our observation that MM-RMs consistently underperform on *o.o.d.* data where text-only shortcuts become ineffective. This observation indicates that samples where unimodal shortcuts fail could potentially be the cases that most require proper multimodal integration. Building upon this insight, we propose to leverage this relationship as a prior knowledge during MM-RM training: **By explicitly identifying and focusing on cases where unimodal spurious correlations fail, we can encourage more robust multimodal reward modeling.** This paradigm shift essentially transforms the curse of text-only shortcuts into an opportunity for improvement, forming the cornerstone of our methodology.

### 4.1. Training with Shortcut-Proxy

We introduce a robust and generalizable algorithm for MM-RM learning. Our goal is to devise a method that can learn generalizable multimodal reward modeling capabilities even when trained on biased datasets, enabling the model to transcend dataset-specific unimodal spurious correlations.

The essence of our methodology is to identify and highlight scenarios where text-only shortcuts fall short. To accomplish this, we propose a dual-branch architecture during the training phase (see Figure 6). Each branch employs an identically initialized reward model but varies in its approach to modality: The primary branch $\mathcal{M}$ is trained on standard multimodal preference data, acting as our Shortcut-aware MM-RM; the auxiliary branch $\mathcal{M}_t$ works with preference data with the image modality removed, serving as a proxy for text-only shortcuts. To quantify and leverage the disparities between the two branches, we introduce the Shortcut-Failure Coefficient (SFC) as follows:

**Definition 4.1.** (Shortcut-Failure Coefficient). This metric measures, from a sample-wise perspective, the proportion of loss contributed by the auxiliary branch (shortcut-proxy) to the total training objective, thus indicating the extent to

which unimodal spurious correlations fail to capture complete preference patterns. Formally, we define SFC as:

$$\text{SFC}(\boldsymbol{x}_i, y_i) = \frac{\mathcal{L}_t(\boldsymbol{x}_i, y_i)}{\mathcal{L}(\boldsymbol{x}_i, y_i) + \mathcal{L}_t(\boldsymbol{x}_i, y_i)} \qquad (5)$$

where

$$\mathcal{L}(\boldsymbol{x}_i, y_i) = -\log(\sigma((r(v_i, q_i, a_i^y) - r(v_i, q_i, a_i^{-y}))))$$
$$\mathcal{L}_t(\boldsymbol{x}_i, y_i) = -\log(\sigma((r_t(q_i, a_i^y) - r_t(q_i, a_i^{-y})))) \qquad (6)$$

Here, $(\boldsymbol{x}_i, y_i)$ is a sample from the training dataset $\mathcal{S}_{train}$, $\mathcal{L}(\boldsymbol{x}_i, y_i)$ and $\mathcal{L}_t(\boldsymbol{x}_i, y_i)$ represent the sample-wise loss values from the primary branch and text-only branch respectively, which are derived from Equations (1) and (3). $r(\cdot)$ and $r_t(\cdot)$ represent the reward functions of the main branch $\mathcal{M}$ and shortcut-proxy $\mathcal{M}_t$ respectively. In practice, we detach the gradients involved in the SFC computation, ensuring that these values only serve as weighting coefficients without participating in backpropagation.

After calculating the SFC values, we can reformulate the loss function of the primary branch into a "shortcut-aware" ($sa$) form as follows:

$$\mathcal{L}_{sa} = \mathbb{E}_{(\boldsymbol{x}_i, y_i) \in \mathcal{S}_{train}} [\text{SFC}(\boldsymbol{x}_i, y_i) \cdot \mathcal{L}(\boldsymbol{x}_i, y_i)] \qquad (7)$$

Essentially, our shortcut-aware loss function utilizes the SFC value to dynamically reweight samples in training distribution: A sample with a higher SFC value indicates that the text-only branch struggles to model preferences, suggesting that multimodal fusion is crucial for robust learning, thus receiving an increased weight; conversely, a sample with a lower SFC value indicates that the text-only branch can easily discriminate it, leading to a reduced weight. **We consider this reweighting mechanism as an adaptive method to shift the training data distribution towards environments where multimodal understanding is crucial.** This approach helps prevent the MM-RM from becoming overly dependent on text-only shortcuts, which have been shown to significantly hinder the model's generalization. We further explore the theoretical underpinnings of our proposed method in Appendix E.

### 4.2. Shortcut-Disentangled Inference

After completing the model training, we can simply remove the auxiliary branch from the architecture, as it only serves as a proxy for text-only shortcuts during training. During inference, we only need to deploy the primary branch, which means the inference process is identical to that of a standard MM-RM, with no additional overhead. However, because the model is trained in a shortcut-aware environment, we demonstrate that it has significantly better generalization and downstream performance (see Section 5).

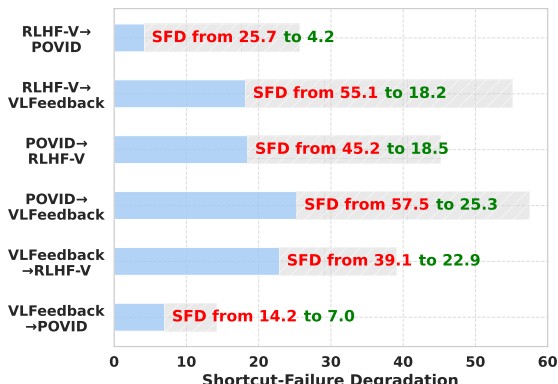

*Figure 7.* Shortcut-Failure Degradation (SFD) of the Shortcut-aware MM-RMs in various *o.o.d.* scenarios. **Compared with standard MM-RMs, Shortcut-aware MM-RMs exhibit consistent robustness improvements.**

While our method explicitly targets text-only shortcuts, the framework's modality-agnostic design allows for broader applicability to other forms of spurious correlations. We discuss this further in Appendix F.

## 5. Experiments

### 5.1. Cross-Distribution Evaluation

This section demonstrates the generalization improvement of our Shortcut-aware MM-RM algorithm under cross-distribution transfer tests. The main settings remain identical to those introduced in Section 3.1, except that we utilize the Shortcut-aware algorithm proposed in Section 4.

As shown in Figure 1(c), compared to standard multimodal RMs, **our Shortcut-aware MM-RMs achieve substantial improvements in generalization performance**, with average accuracy across six *o.o.d.* scenarios increasing from 68.1 to 78.5. We observe that this comes with a slight trade-off in *i.i.d.* performance, where the average accuracy marginally decreases from 91.4 to 90.2.

We further analyze the changes in the Shortcut-Failure Degradation (SFD) metric, which measures the accuracy drop of MM-RMs when text-only features fail in *o.o.d.* scenarios. As shown in Figure 7, our proposed Shortcut-aware algorithm demonstrates robust performance improvements across all *o.o.d.* scenarios, exhibiting substantial and consistent reductions in SFD values compared to standard MM-RMs. This indicates that **Shortcut-aware MM-RMs rely less on text-only shortcuts for reward-scoring and achieve more accurate judgment in scenarios where unimodal spurious correlations fail to generalize.**

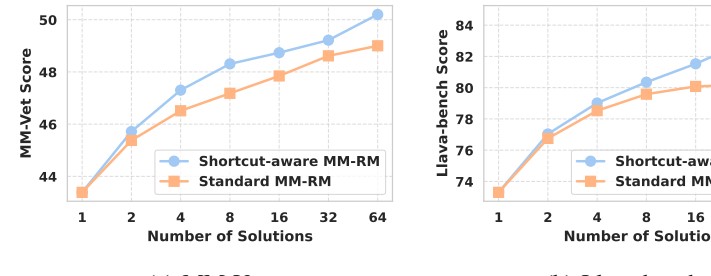 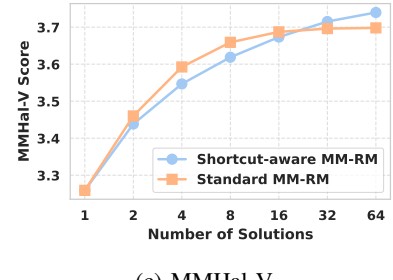

| (a) MM-Vet | (b) Llava-bench | (c) MMHal-V |

*Figure 8.* Best-of-N downstream evaluation with N ranging from 1 to 64, using VLFeedback as the training set. **Shortcut-aware MM-RM demonstrates better scalability, while standard MM-RM shows some degree of reward overoptimization (Gao et al., 2023).** For any N less than 64, we use the unbiased estimator (Nakano et al., 2021) to calculate its average score, as detailed in Appendix H.

**Model Scale Analysis.** To further validate the robustness of our method across different model capacities, we extend experiments to 2B and 4B variants of InternVL2. As shown in Appendix G, the Shortcut-aware algorithm consistently improves generalization on *o.o.d.* scenarios across all model sizes, with average accuracy gains of +6.4 (2B) and +6.8 (4B) compared to standard MM-RMs.

### 5.2. Downstream Test-time Performance

We conduct evaluations on downstream tasks using a Best-of-N (BoN) test-time selection strategy. The process involves generating 64 candidate responses from InternVL2-8B for each image-query pair. MM-RMs are then employed to score these candidates, and the response receiving the highest reward score is selected for downstream evaluation.

To facilitate Best-of-N evaluation, we select three natural QA benchmarks that utilize free-form responses: (1) MM-Vet (Yu et al., 2023b), which comprehensively evaluates VLMs across areas such as visual recognition, OCR, knowledge integration, and other critical capabilities; (2) LLaVA-Bench-in-the-Wild (Liu et al., 2024), which examines the helpfulness of VLMs in aspects like instruction following and complex reasoning; and (3) MMHal-Bench (Sun et al., 2023), which gauges the faithfulness of VLMs by identifying hallucinations in their outputs. To ensure consistency and rigor in the assessment process, we employ GPT-4o (Achiam et al., 2023) as the uniform judge across all three benchmarks. Furthermore, following Amirloo et al., we incorporate images into the MMHal-Bench evaluation process, creating MMHal-V, to enhance assessment accuracy.

The experimental results are presented in Table 2. Notably, **MM-RMs trained with our Shortcut-aware algorithm demonstrate superior Best-of-64 improvements across all benchmarks**, highlighting the algorithm's strong generalization and real-world application value. We also discover that **Shortcut-aware MM-RMs demonstrate better scalability and exhibit stronger robustness against reward**

*Table 2.* Best-of-64 evaluation across downstream tasks, with GPT-4o serving as the universal judge. The first line refers to the results of InternVL2-8B without Best-of-N selection.

| Method | MM-Vet | Llava-bench | MMHal-V |
|---|---|---|---|
| - | 43.4 | 73.3 | 3.26 |
| *MM-RM trained on VLFeedback* | | | |
| Standard | 49.0 | 80.5 | 3.70 |
| Shortcut-aware | **50.2** | **84.7** | **3.74** |
| *MM-RM trained on POVID* | | | |
| Standard | 46.7 | 69.6 | 3.43 |
| Shortcut-aware | **47.3** | **73.8** | **3.71** |
| *MM-RM trained on RLHF-V* | | | |
| Standard | 39.9 | 73.1 | 3.55 |
| Shortcut-aware | **44.5** | **79.6** | **3.61** |

overoptimization (Gao et al., 2023), as shown in Figure 8.

## 6. Related Work

**Reward Modeling.** The development of Large Language Models (Achiam et al., 2023; Bai et al., 2023) heavily relies on Reward Models (Bai et al., 2022), which align models with human values by providing rewards on model responses. An RM is first trained on preference data annotated by humans (Ouyang et al., 2022; Touvron et al., 2023) or AI (Lee et al., 2023; Cui et al., 2024), and can then be incorporated into RLHF (Kaufmann et al., 2023) or rejection sampling (Dubey et al., 2024) during training, or used for response selection (Zhang et al., 2024) and decoding guidance (Khanov et al., 2024) during inference. Recently, with the rise of Multimodal LLMs (Liu et al., 2024; Team et al., 2024), Multimodal RMs (Sun et al., 2023) have gained increasing attention, with works focusing on both the construction (Sun et al., 2023; Wang et al., 2024; Zang et al., 2025) and evaluation (Li et al., 2024a) of MM-RMs.

**Spurious Correlations.** Another line of related work fo-

cuses on spurious correlations (Simon, 1954; Jackson & Somers, 1991; Calude & Longo, 2017), a fundamental challenge that hinders the generalization of intelligent machines. Spurious correlations, also known as shortcut learning (Geirhos et al., 2020; Hermann et al., 2023), occur when certain input features in the training data environments (Arjovsky et al., 2019) are highly correlated with task-specific labels but do not represent the intended task objective. For example, in a cow-camel classification task, if cows are only seen on grassland and camels in deserts during training, the model may rely on background features, failing when animals appear in new contexts (Asgari et al., 2022). Among studies on spurious correlations, the most relevant to our work is unimodal bias (Cadene et al., 2019; Zhang et al., 2023), which examines how multimodal models learn correlations that depend solely on a single modality.

## 7. Conclusion

In this paper, we address a critical challenge of MM-RMs: unimodal spurious correlations that limit their ability to generalize. Our cross-distribution experiments reveal a significant performance disparity of MM-RMs between *i.i.d.* and *o.o.d.* scenarios. Moreover, we find that MM-RMs are able to exploit text-only shortcuts present in multimodal preference datasets, even when trained in a multimodal context, which negatively impacts their generalization. To overcome this limitation, we introduce a Shortcut-aware learning algorithm for MM-RMs, which dynamically identifies and emphasizes samples where text-only shortcuts fail, greatly enhancing their generalization and real-world effectiveness.

## Impact Statement

As Multimodal Large Language Models (MLLMs) gain increasingly diverse means of perceiving the world, aligning them with human values from a multimodal perspective becomes crucial. Multimodal Reward Models (MM-RMs) play a pivotal role in acting as proxies for human preferences. However, research on the generalization capabilities of MM-RMs has been lacking, which poses significant risks. If RMs fail to generalize to out-of-distribution, unseen data, it could lead to MLLMs producing high-reward outputs that do not align with actual human expectations, a phenomenon known as reward hacking, thereby raising concerns about their reliability and safety. By designing a more generalizable MM-RM learning algorithm, our work significantly enhances the generalization capabilities of MM-RMs and their robustness against unimodal spurious correlations. This advancement promotes the reliability of MM-RMs in real-world scenarios, ensuring that MLLMs better adhere to human values, with increased faithfulness and safety, thus having a positive impact on society. Moreover, our proposed method serves as a general algorithm and does not pose any direct negative ethical or social implications.

## Acknowledgements

We sincerely thank the reviewers for their insightful comments and valuable suggestions. This work was supported by Beijing Natural Science Foundation (L243006), Beijing Municipal Science and Technology Project (Nos. Z231100010323002), the Natural Science Foundation of China (No. 62306303).

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

# A. Details of Cross-Distribution Setup

This section mainly supplies detailed setup of the cross-distribution experiments.

## A.1. Data Setup

In this paper, we instantiate the cross-distribution framework proposed in Section 2.2 as a practical experimental setup through the selection of three existing multimodal preference datasets: VLFeedback (Li et al., 2024b), POVID (Zhou et al., 2024), and RLHF-V (Yu et al., 2024). The datasets are carefully partitioned into training and held-out test sets, with strict measures taken to prevent any image overlap between the splits. For the larger VLFeedback dataset, we allocate 1,000 instructions to the test set, while POVID and RLHF-V each receive 400 test instructions.

While each preference dataset focuses on common Vision-Language tasks, such as VQA and image captioning, they derive their instructions from different sources (Yu et al., 2023a; Zhao et al., 2023; Liu et al., 2024). Moreover, they differ in their data annotation and construction paradigms: VLFeedback utilizes AI-generated preference annotations, POVID is created by deliberately introducing hallucinations into standard responses, and RLHF-V employs human labeling to rectify model response errors. This diversity in data sources enhances the representativeness of our experimental findings.

Table 3 summarizes the key characteristics of the three multimodal preference datasets, including their instruction sources, size statistics (instruction and response counts), and annotation methods.

*Table 3.* Three multimodal preference datasets used in cross-distribution experiments.

| Dataset Name | Instruction Source | Instructions *Training* | *Test* | Responses per Instruction | Annotation Paradigm |
|---|---|---|---|---|---|
| VLFeedback | SVIT, LLaVA, etc. | 79,258 | 1,000 | 4 | GPT-4V annotation |
| POVID | LLaVA-Instruct-150k | 16,336 | 400 | 2 | Hallucination Injection |
| RLHF-V | UniMM-Chat | 5,265 | 400 | 2 | Human refinement |

## A.2. Model Setup

We primarily utilize InternVL2-8B (Chen et al., 2024b) as the basis for reward model construction. To adapt the Vision-Lanugage Model to an MM-RM architecture, we remove the language modeling head $l$ and add an RM head $h$, which is a linear layer that converts the hidden states from the last layer of the LLM decoder into a scalar reward score. We implement a dynamic patch mechanism that segments images into up to 6 sub-images for encoding. During the training phase, we froze the vision encoder while only fine-tuning three components: the linear projection module that connects the vision encoder and LLM decoder, the LLM decoder, and the RM head. We train each model variant with its corresponding loss function: standard MM-RM with Equation (1), text-only RM with Equation (3), and our shortcut-aware MM-RM with Equation (7).

We also detail the training hyper-parameters, which remain constant across our experiments, as shown in Table 4.

*Table 4.* Hyper-parameters used in our RM training.

| Hyper-parameter | Value |
|---|---|
| Learning Rate | 1e-5 |
| Global Batch Size | 256 for VLFeedback and POVID, 128 for RLHF-V |
| Training Epoch | 1 |
| Max Input Length | 4096 |
| Optimizer | AdamW |
| Weight Decay | 0.05 |
| Scheduler Type | Cosine |
| Warmup Ratio | 0.1 |

## B. Investigating the Limited Role of Training Data Size in Generalization

In Section 3.2, we present the generalization performance of standard MM-RMs and observe varying generalization capabilities across models trained on different datasets. In this section, we further discuss the role of training dataset size, demonstrating that it is not the decisive factor in MM-RMs' generalization.

Since VLFeedback has the best generalization ability (which also has the largest data scale), we randomly downsample the VLFeedback training data to eliminate its scale advantage in instruction data size compared to the other two training datasets. However, we find that MM-RMs built on the downsampled VLFeedback training data still exhibit better generalization abilities compared to models built on other datasets (see Table 5). This suggests that the size of the training data plays only a limited role in the model's generalization ability and is not a decisive factor.

We believe that the varying degrees of unimodal spurious correlations in different datasets are a more critical reason that prevents MM-RMs from generalizing effectively. This is one of the key motivations for introducing the Shortcut-aware algorithm in our paper, which is effective on training data of different sizes and with varied unimodal spurious correlations, thereby comprehensively enhancing the generalization ability of MM-RMs.

*Table 5.* Impact of VLFeedback training data size on the standard MM-RM performance.

| Training Set | | Test Set | | | Performance Comparison | |
| --- | --- | --- | --- | --- | --- | --- |
| *Dataset* | *Instructions* | *VLFeedback* | *POVID* | *RLHF-V* | *in-distribution* | *out-of-distribution* |
| | 3k | 86.3 | 90.3 | 67.3 | 86.3 | 78.8 |
| VLFeedback | 5k | 86.4 | 88.5 | 68.3 | 86.4 | 78.4 |
| | 79k | **87.1** | 89.5 | 69.0 | 87.1 | 79.3 |
| POVID | 16k | 60.0 | **100.0** | 47.8 | 100.0 | 53.9 |
| RLHF-V | 5k | 67.5 | 75.0 | **87.0** | 87.0 | 71.3 |

## C. Further Discussion on Shortcut-Failure Degradation

The Shortcut-Failure Degradation (SFD) metric serves as a specialized diagnostic tool for analyzing unimodal spurious correlations in MM-RMs. By measuring the accuracy gap between shortcut-success and shortcut-fail subsets, SFD directly quantifies a model's dependence on text-only shortcuts. This makes it particularly valuable for identifying unimodal bias, as it isolates this specific failure mode from other potential robustness issues that might be captured by more general metrics.

Conceptually, SFD aligns with established robustness evaluation paradigms like worst-group accuracy (Sagawa et al., 2019; Nam et al., 2022; Chaudhuri et al., 2023), where the shortcut-fail subset represents our "worst group" in multimodal generalization. Both approaches focus on model performance in challenging scenarios where biased correlations break down. Just as worst-group accuracy reveals models that exploit majority-group features, SFD exposes models that over-rely on text shortcuts that may dominate training but fail during deployment.

From a practical standpoint, SFD's significance becomes particularly clear in real-world applications. For instance, an MM-RM might learn to favor "long, descriptive captions" as a text shortcut during training, but would fail dramatically when deployed to domains where response quality critically depends on visual understanding (e.g., medical imaging or technical diagram interpretation). SFD precisely measures this vulnerability by evaluating performance when text shortcuts become unreliable, making it an essential metric for assessing MM-RM robustness in practical deployment scenarios.

## D. Investigating the Impact of Feature Acquisition Difficulty on Generalization

In this section, we conduct initial research to explore whether the limited representation capacity causes MM-RMs to prioritize the learning of unimodal spurious correlations under empirical risk minimization (Vapnik, 1991; Yang et al., 2022; Hermann et al., 2023). We refer to the text-only shortcuts as "easily acquired features," which stem from the modality gap (Liang et al., 2022): A fixed-size image can encompass an abundance of fine-grained details, including shapes, textures, and backgrounds, leading to a significant information imbalance when compared to the textual content (Schrodi et al., 2024).

Specifically, we conduct scaling experiments by using the maximum number of visual patches as an indicator of feature

acquisition difficulty, which represents the upper limit of sub-images derived from image segmentation (see Figure 9). A larger number of visual patches enables the reward model to capture more fine-grained visual features.

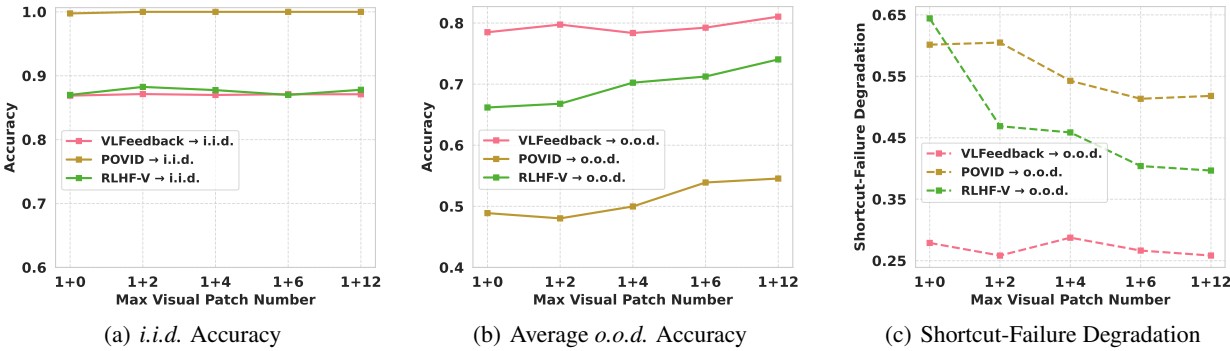

(a) *i.i.d.* Accuracy    (b) Average *o.o.d.* Accuracy    (c) Shortcut-Failure Degradation

*Figure 9.* Impact of max number of visual patches on standard MM-RM performance.

We observe that: in *i.i.d.* test scenarios, the model's performance remains stable regardless of the number of visual patches. In contrast, under *o.o.d.* test scenarios, we find that providing fine-grained visual features enhances the generalization capability of MM-RMs. Except for models trained on VLFeedback, which already demonstrate relatively strong generalization, we observe an overall increase in accuracy and decrease in SFD values. This suggests that adjusting the accessibility of visual features can help reduce the learning of text-only shortcuts.

However, current results indicate that simply increasing the number of visual patches cannot systematically address the model's robustness issues, as the improvement in generalization is not substantial, and the SFD values barely decrease when scaling the number of visual patches from 6 to 12. While future multimodal architectures that provide more accessible visual features could be a promising direction for mitigating unimodal shortcuts, we propose in this paper an effective shortcut-invariant MM-RM algorithm that is applicable to all architectures, as introduced in Section 4.

## E. Further Discussion on the Proposed Method

In this section, we further discuss how our Shortcut-aware MM-RM learning algorithm provides both an information-theoretic justification for dynamic sample weighting and a practical connection to robust optimization, ensuring that MM-RMs prioritize learning truly multimodal reward functions.

**Information-Theoretic Interpretation of SFC.** The Shortcut-Failure Coefficient (SFC) can be understood information-theoretically as quantifying the insufficiency of text-only shortcuts for prediction. Assuming optimized losses approximate conditional entropy terms, we can relate SFC to the underlying information structure:

$$\text{SFC} = \frac{\mathcal{L}_t}{\mathcal{L}_t + \mathcal{L}_m} \approx \frac{H(Y|T)}{H(Y|T) + H(Y|T, V)} \tag{8}$$

Defining $\rho = \frac{I(V, Y|T)}{H(Y|T)}$ as the normalized contribution of visual information beyond text, we derive that SFC $\approx \frac{1}{2-\rho}$. This shows that SFC monotonically increases with $\rho$, meaning samples requiring stronger visual grounding receive higher weights. Thus, the Shortcut-Failure Coefficient provides a principled foundation for reweighting based on the unique predictive contribution of multimodal information.

**Connection to Distributionally Robust Optimization.** Our method also shares conceptual connection with Group Distributionally Robust Optimization (Sagawa et al., 2019), , with each sample representing a group without explicit labels. We hypothesize that worst-case group probability depends on the degree to which samples cannot be predicted using text information alone, aligning with our observations of Shortcut-Failure Degradation.

By setting sample weights as: $w(\boldsymbol{x}_i, y_i) = \frac{\text{SFC}(\boldsymbol{x}_i, y_i)}{\mathbb{E}_{(\boldsymbol{x}_i, y_i)}[\text{SFC}(\boldsymbol{x}_i, y_i)]}$ , we effectively shift the training distribution from $\mathbb{P}$ to $\mathbb{Q}$, where: $\frac{d\mathbb{Q}}{d\mathbb{P}}(\boldsymbol{x}_i, y_i) \propto w(\boldsymbol{x}_i, y_i)$ . These weights positively correlate with the optimal reweighting for minimizing worst-case

risk, as they upweight samples where unimodal shortcuts are inadequate. This also aligns with our empirical observations in Section 5.1, where SFC-based reweighting significantly reduces generalization gaps.

## F. Adaption to Other Spurious Correlations

While our method explicitly targets text-only shortcuts, the framework's modality-agnostic design allows for broader applicability to other forms of spurious correlations. To adapt the algorithm, two key modifications are required: **proxy adaptation** and **dynamic reweighting**. First, the auxiliary branch can be replaced with a problem-specific proxy (e.g., an image-only branch) to isolate the target spurious feature. Second, analogous to the Shortcut-Failure Coefficient (SFC) in Equation 5, failure signals from the new proxy can be computed to reweight training samples where the proxy struggles, thereby prioritizing cases requiring robust learning.

This flexibility stems from the core principle of our approach: dynamically shifting focus toward samples where spurious correlations fail, regardless of their modality. For instance, if background biases dominate a dataset (e.g., visual QA), training a background-only proxy and reweighting based on its failure rate would similarly encourage reliance on complementary information. The reweighting mechanism remains unchanged, ensuring the framework's scalability while preserving its effectiveness in mitigating diverse spurious correlations.

## G. Model Size Effects on Cross-Distribution Performance

To investigate the impact of model capacity on generalization, we extend our experiments to 2B, 4B of the base architecture InternVL, as shown in Figure 10. While increasing model scale improves overall performance—with 8B achieving higher absolute accuracy than smaller models—we observe that scaling alone does not fully resolve unimodal shortcut reliance. For example, in challenging out-of-distribution scenarios (e.g., POVID → RLHF-V), all model scales exhibit poor performance, demonstrating that merely increasing model parameter scale is insufficient to completely address generalization issues.

Critically, our shortcut-aware algorithm consistently improves generalization across all model sizes, while demonstrating considerable scalability (see Figure 11). In out-of-distribution scenarios, the 2B, 4B, and 8B models achieve average accuracy gains of +6.4, +6.8, and +10.4 respectively, compared to standard MM-RMs.

This demonstrates that our dynamic reweighting method addresses fundamental limitations in multimodal alignment—complementing (rather than being superseded by) increased capacity. While scaling plays a certain role, explicit shortcut mitigation remains essential for robust reward modeling.

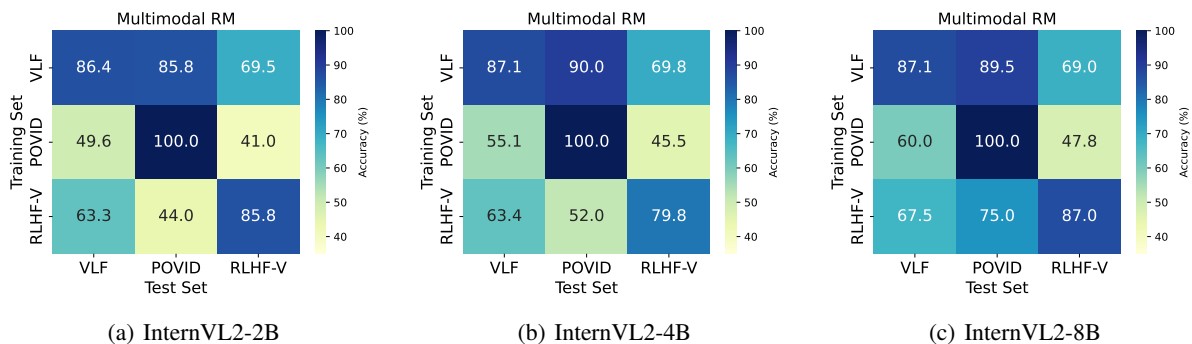

*Figure 10.* Cross-distribution evaluation of standard MM-RM across different model scales.

## H. Best-of-N Unbiased Estimator

Following previous work (Nakano et al., 2021; Coste et al., 2023), we use an unbiased estimator to calculate the average score of the Best-of-N policy for any $N < 64$, where 64 represents the total number of sampled responses. This approach enables more reliable evaluation. Specifically, the estimator computation is as follows:

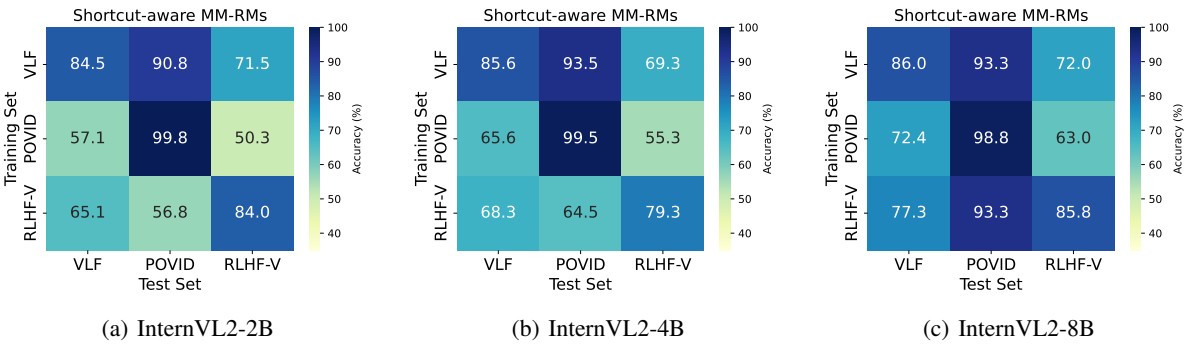

*Figure 11.* Cross-distribution evaluation of our Shortcut-aware MM-RM across different model scales.

$$S_{\text{Best-of-}N}(v, q) = \frac{1}{\binom{64}{N}} \sum_{1 \leq i_1 < \cdots < i_N \leq 64} S^{\text{GPT-4o}} \left( \underset{a \in \{A_{i_1}, \cdots, A_{i_N}\}}{\arg\max} \, r(v, q, a) \right) \tag{9}$$

Here, $r(\cdot)$ represents the reward function of the evaluated MM-RM; $(v, q)$ denotes a vision-related query from a downstream benchmark; $a$ represents an answer sampled from the policy model (InternVL2-8B (Chen et al., 2024b)); and $S^{\text{GPT-4o}}(a)$ indicates the score of answer $a$ evaluated by the judge GPT-4o (Achiam et al., 2023).

