# OpenReview forum: "The Devil Is in the Details: Tackling Unimodal Spurious Correlations for Generalizable Multimodal Reward Models"
_ICML.cc/2025/Conference — ICML 2025 poster_

### Official Review · Reviewer_Rc2L · 2025-03-11

**Overall Recommendation:** 3

**Summary:**

The paper addresses the challenge of unimodal spurious correlations in Multimodal Reward Models (MM-RMs), particularly how these models fail to generalize to out-of-distribution (o.o.d.) data.  These spurious correlations occur when models over-rely on text-only features, which hold in the training distribution but fail in o.o.d. settings.  The authors introduce a “Shortcut-aware” MM-RM learning algorithm that dynamically identifies and reweights samples where unimodal shortcuts fail, thus promoting a stronger multimodal understanding.  Experimental results show improvements in generalization and performance across various tasks, suggesting that this approach is effective in reducing the impact of unimodal biases.

**Claims And Evidence:**

The claims about unimodal spurious correlations limiting MM-RMs’ generalization to o.o.d. data are supported by experimental evidence, particularly the cross-distribution tests.  However, the discussion could benefit from more detailed analysis on the theoretical aspects of why the proposed approach outperforms other methods, especially in terms of sample reweighting.  The evidence is generally convincing, but the connection between the empirical findings and the theoretical foundations could be made clearer.

**Essential References Not Discussed:**

No.

**Experimental Designs Or Analyses:**

The experimental setup appears solid, with a clear focus on cross-distribution and real-world applicability.  The use of multiple datasets (VLFeedback, POVID, RLHF-V) strengthens the claims about generalization.

**Methods And Evaluation Criteria:**

The proposed methods, specifically the Shortcut-aware learning algorithm and the use of the Shortcut-Failure Coefficient (SFC), are appropriate for the problem of improving generalization in multimodal models.   The experimental design, including cross-distribution and downstream performance evaluations, is well-suited to test the algorithm’s robustness.   However, while the evaluation metrics like SFD (Shortcut-Failure Degradation) are useful, they might need further justification in terms of how they specifically relate to the real-world applicability of MM-RMs.

**Other Comments Or Suggestions:**

The paper is well-written but could improve in terms of theoretical clarity, especially concerning the Shortcut-aware algorithm’s formal underpinnings.  It might also be helpful to include more experimental results on how the algorithm performs across a broader range of real-world scenarios, particularly in domains outside of vision-language models.

**Other Strengths And Weaknesses:**

A key strength of the paper is its novelty in tackling unimodal spurious correlations in multimodal models.  The Shortcut-aware algorithm is a promising contribution that addresses a clear gap in the current research.  However, the paper could be clearer in its discussion of how the proposed method compares quantitatively to other established methods for multimodal model generalization.  The evaluation could also benefit from more rigorous analysis of the trade-offs between in-distribution and out-of-distribution performance.

**Questions For Authors:**

1.  How do you justify the use of the Shortcut-Failure Coefficient (SFC) as a meaningful metric for measuring the failure of unimodal shortcuts?
2.  Could you provide a more detailed theoretical analysis of how the Shortcut-aware algorithm ensures better generalization in multimodal settings?  Would other debiasing techniques, such as invariant risk minimization, perform similarly in your setup?

**Relation To Broader Scientific Literature:**

The work is well-grounded in the current literature on reward models and multimodal learning.   The authors effectively position their work within the context of previous research on unimodal biases and generalization in MM-RMs.

**Theoretical Claims:**

The paper introduces some theoretical concepts like the Shortcut-Failure Coefficient (SFC), but does not provide a full theoretical proof of how this approach leads to better generalization.  This is an area that could use more depth, particularly regarding how the shortcut-aware approach compares to standard training regimes in a formal theoretical context.

---

> ### Author Rebuttal · Authors · 2025-04-01
>
> Thank you for your recognition of the strengths of our work. We address your concerns as follows.
>
> **C1: More detailed analysis on the theoretical aspects of why the proposed approach outperforms other methods.**
>
> **C2: Justify the use of SFC as a meaningful metric for measuring the failure of unimodal shortcuts.**
>
> A1-2: Thank you for your interest in our theoretical analysis. Below is a more detailed justification:
> - Information-theoretically, SFC quantifies text-only shortcut insufficiency for prediction. SFC = $\frac{L_t}{L_t + L_m}$ estimates $\frac{H(Y|T)}{H(Y|T) + H(Y|T,V)}$ assuming optimized losses approximate entropy terms. Defining $\rho = \frac{I(V,Y|T)}{H(Y|T)}$ as visual modality's normalized information contribution, we derive: $SFC \approx \frac{1}{2-\rho}$. This establishes SFC as monotonically increasing with $\rho$, providing principled foundation for reweighting based on visual information's unique contribution.
> - Our method connects with group DRO [1], with each sample representing a group without explicit labels. We hypothesize that worst-case group probability depends on the degree to which samples cannot be predicted using text information alone, aligning with our observations of Shortcut-Failure Degradation. By defining weights as $w(x, y) = \frac{SFC(x, y)}{\mathbb{E}{(x, y)} [SFC(x, y)]}$, we shift from $P_{train}$ to $Q$ where $\frac{dQ}{dP_{train}}(x, y) \propto w(x, y)$. These weights positively correlate with optimal weights minimizing worst-case risk, reducing generalization gaps while maintaining computation efficiency.
>
> We will expand this analysis in the future revision.
>
> **C3: While the evaluation metrics like SFD are useful, they might need further justification in terms of how they specifically relate to the real-world applicability of MM-RMs.**
>
> A3: Thank you for this important point about SFD's real-world relevance. SFD quantifies MM-RM performance when text shortcuts fail - a scenario critical for practical applications. Consider a concrete example: an MM-RM might learn to prefer "long, descriptive captions" as a text shortcut, but would fail when deployed to a domain where caption quality depends on visual grounding (e.g., medical imaging). SFD measures how severely the MM-RM fails in such scenarios. We'll add more discussion about SFD's practical implications in our next revision.
>
> **C4: The evaluation could also benefit from more rigorous analysis of the trade-offs between in-distribution and out-of-distribution performance.**
>
> A4: We appreciate your insightful comments. Our experiments reveal a nuanced balance: while the proposed algorithm slightly reduces IID accuracy (91.4→90.2), it achieves an absolute gain in OOD scenarios (68.1→78.5)—a critical robustness improvement. This trade-off aligns with our expectations: suppressing text-only shortcuts weakens overfitting to dataset-specific biases, which marginally impacts IID performance but significantly enhances generalization. We will provide more detailed discussion in our revision.
>
> **C5: The paper could be clearer in its discussion of how the proposed method compares quantitatively to other established methods for multimodal model generalization.**
>
> A5: Thank you for your valuable suggestion. We would like to clarify that the research landscape shows an imbalance: text-only RM generalization are well-studied [2] while MM-RMs remain under-explored despite their growing importance. Our work addresses this gap by identifying and mitigating unimodal spurious correlations that significantly impair MM-RM generalization, establishing new understanding in this emerging area.
>
> **C6: Would other debiasing techniques, such as invariant risk minimization, perform similarly in your setup?**
>
> A6: While IRM and related work inspire us, we believe that direct application of IRM would be infeasible in our setup:
> 1) IRM requires multiple training environments with explicit distribution shifts. In contrast, real-world RMs are typically trained on a single environment due to practical constraints in data collection [3]. Therefore, our setup limits the training data to a single environment, while our method circumvents this by implicitly defining "environments" per sample via SFC.
> 2) IRM and similar algorithms involve bi-level optimization with second-order gradients, challenging to optimize for large models like InternVL2-8B due to computational overhead and optimization instability. Our approach instead builds on ERM with sample-wise risk reweighting, ensuring compatibility with existing training frameworks and efficient backpropagation.
>
> [1] Sagawa, Shiori, et al. "Distributionally robust neural networks for group shifts: On the importance of regularization for worst-case generalization." ICLR, 2020.
>
> [2] Yang, Rui, et al. "Regularizing hidden states enables learning generalizable reward model for llms." NeurIPS 2024.
>
> [3] Ouyang, Long, et al. "Training language models to follow instructions with human feedback." NeurIPS 2022.

---

### Official Review · Reviewer_hs7o · 2025-03-13

**Overall Recommendation:** 3

**Summary:**

This paper highlights how unimodal spurious correlations reduce generalization in multimodal reward models. In cross-distribution tests, MM-RMs trained on large, seemingly robust datasets still fail to generalize in unseen environments, primarily because they exploit textual cues rather than genuinely integrating vision and language. The paper proposes a “Shortcut-aware” algorithm that dynamically weights training examples where text-only reasoning fails, thereby forcing the model to rely more on multimodal inputs. Empirical results show that this method improves out-of-distribution accuracy while modestly sacrificing in-distribution performance, leading to better overall robustness and downstream task performance.

**Claims And Evidence:**

Yes

**Essential References Not Discussed:**

NA

**Experimental Designs Or Analyses:**

Yes

**Methods And Evaluation Criteria:**

Yes

**Other Comments Or Suggestions:**

NA

**Other Strengths And Weaknesses:**

Strengths:
1. The paper identifies a very practical gap in multimodal reward modeling: many so-called multimodal models mainly leverage textual signals, limiting their reliability on unseen data.

2. The proposed dynamic reweighting mechanism, which highlights cases where text-only responses fail, is conceptually straightforward but shows notable gains in generalization, a good example of tackling “shortcut learning” with minimal changes to the overall architecture.

Weaknesses:
1. While the paper presents a creative fix, it does not introduce new core ML techniques or theory. The reweighting approach, although usefu, mostly builds on known dynamic data debiasing heuristics.

2. There is little formal treatment of how or why the method shifts model capacity away from text-only cues toward genuinely multimodal representations. This lack of theoretical grounding might limit the paper’s appeal.

3. The problem is demonstrated in the context of MM-RMs for reward modeling. Extensions to tasks like sequence generation, multi-turn conversation, or reinforcement learning from (truly) diverse signals remain unexplored.

4. Although the empirical results are valuable, the paper’s novelty is mainly spotting a subtle problem in multimodal RMs and offering a practical fix. From a rigorous ML research perspective, it may be viewed more as an engineering improvement rather than breaking fresh ground in ML methodology.

**Questions For Authors:**

See Weaknesses Above.

**Relation To Broader Scientific Literature:**

NA

**Theoretical Claims:**

No

---

> ### Author Rebuttal · Authors · 2025-04-01
>
> Thank you for your recognition of the strengths of our work. We address your concerns as follows.
>
> **C1: While the paper presents a creative fix, it does not introduce new core ML techniques or theory.**
>
> **C2: From a rigorous ML research perspective, it may be viewed more as an engineering improvement rather than breaking fresh ground in ML methodology.**
>
> A1-2: We sincerely thank you for recognizing our method’s creativity and for the thoughtful critique. We appreciate the chance to clarify our contributions.
>
> - We systematically reveal unimodal spurious correlations as a critical bottleneck in MM-RM generalization - a novel insight in multimodal alignment. While prior works address single-modality biases (e.g., text-length bias [1]), we uniquely demonstrate how modality gaps fundamentally drive models toward text-only shortcuts, even in multimodal training paradigms. This discovery exposes a critical blind spot in existing approaches and establishes a new research direction for multimodal robustness.
> - We propose a principled framework anchored in the Shortcut-Failure Coefficient (SFC), which dynamically quantifies modality reliance through dual-branch interaction. Unlike heuristic reweighting methods, SFC leverages intrinsic modality conflicts to identify samples requiring multimodal integration. Our cross-distribution evaluation protocol and Shortcut-Failure Degradation metric further provide standardized tools for diagnosing generalization failures, enabling systematic comparisons in future work.
> - We conduct rigorous experiments to analyze the impact of text-only shortcuts on generalization and demonstrate the real-world effectiveness of our approach. Our method not only achieves significant generalization gains (e.g., +10.4 average OOD accuracy) but also improves performance on real-world downstream tasks (e.g., +6.5 on LLaVA-Bench). These results underscore that our approach is not just a heuristic but a principled framework toward more robust multimodal systems.
>
> While building on existing ML components, our work establishes a new paradigm for analyzing and mitigating modality-specific biases - a fundamental challenge as multimodal systems scale. We believe these contributions meaningfully advance the development of robust multimodal alignment.
>
> **C3: There is little formal treatment of how or why the method shifts model capacity away from text-only cues toward genuinely multimodal representations. This lack of theoretical grounding might limit the paper’s appeal.**
>
> A3: Thank you for your interest in the theoretical analysis of our approach. Due to space limitations, we follow ICML’s guidelines and recommend that you refer to our response to Reviewer Rc2L, where we provide a theoretical analysis of how our SFC measure quantifies the insufficiency of text-only shortcuts for prediction from an information-theoretic perspective. Additionally, we analyze how sample reweighting can shift the training distribution to optimize worst-case OOD risk. This theoretical foundation illustrates how our method reallocates model capacity away from text-only cues toward genuinely multimodal representations.
>
> **C4: The problem is demonstrated in the context of MM-RMs for reward modeling. Extensions to tasks like sequence generation, multi-turn conversation, or reinforcement learning from (truly) diverse signals remain unexplored.**
>
> A4: We appreciate the reviewer's concern about the scope of our work. We would like to clarify that while our experiments focus on MM-RMs as the implementation context, the implications and applications of our findings are much broader.
>
> Reward models serve as preference modeling tools that guide AI responses across various paradigms. As essential components shaping AI behavior in both training and inference stages, they directly connect to tasks including the sequence generation, conversation, and RL applications the reviewer mentioned:
> 1. Our experiments demonstrate how MM-RMs improve model outputs through Best-of-N selection at test time across diverse benchmarks including MM-Vet, Llava-Bench, and MMHal-V. These benchmarks specifically evaluate capabilities in visual recognition, OCR, instruction following, and complex reasoning - all essential components of effective sequence generation and conversational AI.
> 2. We have supplemented our initial experiments with additional results showing how our improved MM-RMs enhance policy models when used in RL training loops. The policy models trained with our shortcut-aware MM-RMs demonstrate significant performance improvements across the same benchmarks.
> ||MM-Vet|Llava-bench|MMHal-V|
> |-|-|-|-|
> |Policy (InternVL2-8B)|43.4|73.3|3.26|
> |Policy + RL (w/ Standard MM-RM)|43.5|76.3|3.46|
> |Policy + RL (w/ Shortcut-aware MM-RM)|**44.1**|**77.8**|**3.58**|
>
> Please let us know if you have any additional questions.
>
> [1] Chen, Lichang, et al. "Odin: Disentangled reward mitigates hacking in rlhf." ICML 2024.

---

### Official Review · Reviewer_6qLS · 2025-03-14

**Overall Recommendation:** 3

**Summary:**

This paper reveals that multimodal reward models (MM-RMs) are struggling to address out-of-distribution (OOD) input queries and identify the unimodal spurious correlation (text-only reliance behavior of MM-RMs) as the main cause of this issue. The authors provide some hypotheses behind this issue and propose to leverage this bias to dynamically reweight the per-sample loss based on the expertise of text-only and multimodal reward models. This shortcut-aware multimodal reward modeling induces better OOD generalization of the reward model and improves downstream open-ended question-answering tasks through best-of-N inference protocol.

**Claims And Evidence:**

The main claim is that the poor OOD generalization capability of MM-RMs is due to the text-only shortcut, and this claim was validated through systemic empirical evaluation.

**Essential References Not Discussed:**

The authors properly cited essential references.

**Experimental Designs Or Analyses:**

The experiments are intuitively designed and valid to investigate the authors' hypotheses, but there is a slight concern about the volume of the experiment (which will be covered in the strengths and weaknesses section).

**Methods And Evaluation Criteria:**

The proposed method and evaluation protocol are both reasonable enough to make a persuasive empirical validation.

**Other Comments Or Suggestions:**

* I would recommend the author to remove the negative sign from equation (6) and equation (7). In its current form, the statement "A sample with a higher SFC value indicates that the text-only branch struggles to model preferences, suggesting that multimodal fusion is crucial for robust learning, thus receiving an increased weight ..." is wrong because the higher value of SFC means text-only branch produces large preference loss (the logarithm of sigmoid is range from -inf to 0). If the negative signs in Eq (6) and (7) are removed, the above explanation will be correct.
* It would be great if the authors could further provide some justifications for the proposed metric Shortcut-Failure Degradation (SFD). Is it correlated with other robustness metrics? or is it just for measuring the degree of text-only shortcut bias inside of MM-RMs?

# Post-rebuttal
> The authors' rebuttal is quite professional and addresses some of my concerns successfully. I will be thinking about editing my initial review and rating after carefully going through other rebuttal contents to other reviewers (but will not require any additional details or raise questions from/to authors). -> After looking through the other reviews and rebuttals, I want to adhere to my current rating -- Weak Accept -- by echoing the concerns raised by `hs7o ` while acknowledging some unique insights provided by this work.

**Other Strengths And Weaknesses:**

## Strengths
* The presentation quality is excellent.
* The main hypothesis -- MM-RMs' poor OOD robustness is due to text-only shortcuts -- is well-supported by extensive intermediate experiments.
* The authors try to further dive into the reason behind the phenomenon (text-only shortcuts) by providing some conjectures (information imbalance between modalities and so on).

## Weaknesses
* Although the authors provide rich analysis to strengthen the motivation and rationale behind the method development, one concern is that they only validate their approach with InternVL2-8B. It is not sure that the proposed method can be applied to other model architectures / model sizes.
* The proposed method hosts an additional large language model branch to model the text-only shortcut bias. This will significantly increase the amount of resources required during reward modeling, and the authors do not provide cost comparison between standard reward modeling and their proposal.

**Questions For Authors:**

See the above reviews.

**Relation To Broader Scientific Literature:**

There have been many attempts to leverage biases (by modeling them with an auxiliary model) to mitigate biases from the main branch network [Bahng et al. 2020; Jung et al. 2023]. This work presents a simple yet effective application of this kind of approaches in the multimodal reward modeling regime.


- [Bahng et al. 2020] Learning De-biased Representations with Biased Representations
- [Jung et al. 2023] Fighting Fire with Fire: Contrastive Debiasing without Bias-free Data via Generative Bias-transformation

**Theoretical Claims:**

There are no theorems.

---

> ### Author Rebuttal · Authors · 2025-04-01
>
> Thank you for your recognition of the strengths of our work. We address your concerns as follows.
>
> **C1: Although the authors provide rich analysis to strengthen the motivation and rationale behind the method development, one concern is that they only validate their approach with InternVL2-8B. It is not sure that the proposed method can be applied to other model architectures / model sizes.**
>
> A1: We appreciate the reviewer's concern about the broader applicability of our method. To address this point, we have conducted additional experiments on other base models:
>
> 1. InternVL2-2B (Model Backbone: InternLM2-Chat-1.8B)
> |Training Set|IID Accuracy ($\Delta$)| OOD Accuracy ($\Delta$)|
> |-|-|-|
> |VLFeedback|84.5 (-1.9)|81.1 (+3.5)|
> |POVID|99.8 (-0.3)|53.7 (+8.4)|
> |RLHF-V|84.0 (-1.8)|60.9 (+7.3)|
>
> 2. InternVL2-4B (Model Backbone: Phi-3-Mini-128K-Instruct)
> |Training Set|IID Accuracy $\Delta$| OOD Accuracy $\Delta$|
> |-|-|-|
> |VLFeedback|85.6 (-1.5)|81.4 (+1.5)|
> |POVID|99.5 (-0.5)|60.4 (+10.1)|
> |RLHF-V|79.3 (-0.6)|66.4 (+8.7)|
>
> $\Delta$ measures the accuracy differential between Shortcut-aware MM-RM and Standard MM-RM. The results demonstrate that our method generalizes well across different model architectures and parameter scales, confirming that the effectiveness of our approach extends beyond just InternVL2-8B.
>
> **C2: The proposed method hosts an additional large language model branch to model the text-only shortcut bias. This will significantly increase the amount of resources required during reward modeling, and the authors do not provide cost comparison between standard reward modeling and their proposal.**
>
> A2: Thank you for raising this important concern about computational resources. We would like to clarify two key points:
>
> 1. The additional text-only branch is only present during the training phase. During inference, this branch is completely removed, meaning our approach has identical computational requirements to standard multimodal reward modeling when deployed in practice.
> 2. The additional computational overhead is comparable to what's already widely accepted in language model alignment research (similar to DPO [1] algorithms that also use a reference model). In the context of the overall model development pipeline (especially compared with pre-training stage), this marginal increase is quite modest.
>
> **C3: I would recommend the author to remove the negative sign from equation (6) and equation (7).**
>
> A3: Thank you for your thorough review of our paper—your attention to detail is valuable for improving our work! We apologize for the typo error you identified. We will correct this mistake in future versions of the paper to ensure mathematical consistency with our textual statement.
>
> **C4: It would be great if the authors could further provide some justifications for the proposed metric Shortcut-Failure Degradation (SFD). Is it correlated with other robustness metrics? or is it just for measuring the degree of text-only shortcut bias inside of MM-RMs?**
>
> A4: Thank you for raising this important question about the justification of our SFD metric. We would like to clarify that:
>
> - SFD addresses the specific challenge of unimodal spurious correlations in MM-RMs. By calculating the accuracy difference between shortcut-success samples and shortcut-fail samples, SFD directly quantifies how much a multimodal model relies on text-only shortcuts. This makes it a tailored metric for diagnosing unimodal bias, unlike general robustness metrics that might conflate multiple failure modes.
> - While SFD isn't directly equivalent to existing metrics, it shares conceptual grounding with worst-group accuracy [2][3][4], where the shortcut-fail subset represents our "worst group" in the multimodal generalization context. Both metrics prioritize performance on challenging subgroups where biased correlations lead to failures. Just as worst-group accuracy penalizes models exploiting majority-group features, SFD penalizes over-reliance on text shortcuts that may dominate training but fail during test phase.
>
> We will elaborate on these connections in our revision.
>
> **C5: There have been many attempts to leverage biases to mitigate biases from the main branch network [Bahng et al. 2020; Jung et al. 2023].**
>
> A5: Thank you for pointing out the pioneering papers related to our research direction! We will cite them in the future version of our paper.
>
> Please let us know if you have any additional questions.
>
> [1] Rafailov, Rafael, et al. "Direct preference optimization: Your language model is secretly a reward model." NeurIPS, 2023.
>
> [2] Sagawa, Shiori, et al. "Distributionally robust neural networks for group shifts: On the importance of regularization for worst-case generalization." ICLR, 2020.
>
> [3] Nam J, Kim J, Lee J, et al. "Spread Spurious Attribute: Improving Worst-group Accuracy with Spurious Attribute Estimation." ICLR, 2022.
>
> [4] Chaudhuri, Kamalika, et al. "Why does throwing away data improve worst-group error?." ICML, 2023.

---

### Official Review · Reviewer_Dzcf · 2025-03-21

**Overall Recommendation:** 3

**Summary:**

The paper proposes to improve the generalization of Multimodal Reward Models (MM-RMs) by addressing the issue of unimodal spurious correlations. It introduces a Shortcut-aware MM-RM learning algorithm that dynamically reweights training samples to emphasize multimodal understanding, reducing reliance on text-only shortcuts. Experiments show the effectiveness in generalization, downstream task performance, and scalability.

**Claims And Evidence:**

The authors demonstrate through cross-distribution transfer experiments that standard MM-RMs struggle with generalization due to unimodal spurious correlations. They introduce the Shortcut-Failure Degradation (SFD) metric to quantify this issue and show that their proposed algorithm reduces SFD values, indicating better generalization. However, it would be more demonstrative if the authors could conduct evaluations on more benchmarks like MMMU and MMStar.

**Essential References Not Discussed:**

N/A

**Experimental Designs Or Analyses:**

All experiments are checked. For the results on Table 2, have the authors tried to combine these three training datasets to train the MM-RM? Would combining preference datasets obtained through different construction methods help, or would they interfere with each other?

**Methods And Evaluation Criteria:**

Yes

**Other Comments Or Suggestions:**

It would be helpful to include a comparison with other state-of-the-art methods for improving generalization in multimodal reward models.

**Other Strengths And Weaknesses:**

- Strengths:  The introduction of the SFD metric provides a new way to diagnose and measure the impact of unimodal spurious correlations.
- Weaknesses: The paper focuses on a specific type of spurious correlation (text-only shortcuts) and may not address other forms of spurious correlations that could exist in multimodal data. Additionally, the method relies on the availability of multimodal preference datasets, which may not always be practical in real-world applications.

**Questions For Authors:**

- Can the Shortcut-aware learning algorithm be extended to address other forms of spurious correlations beyond text-only shortcuts? If so, what modifications would be required?
- How does the method handle scenarios where multimodal preference datasets are limited or unavailable? Are there any alternative strategies to ensure robust generalization in such cases?
- Could we potentially avoid phenomena like text-only shortcuts by better constructing the Preference dataset? Or could this improve the generalization performance of the MM-RMs trained with it?

**Relation To Broader Scientific Literature:**

This paper attempts to addresses a critical limitation of existing MM-RMs by proposing a novel learning algorithm that enhances generalization. A more-robust MM-RMs can enhance the alignment of MLLMs.

**Theoretical Claims:**

The paper does not present any formal theoretical proofs. All equations and definitions are checked.

---

> ### Author Rebuttal · Authors · 2025-04-01
>
> Thank you for your recognition of the strengths of our work. We address your concerns as follows.
>
> **C1: It would be more demonstrative if the authors could conduct evaluations on more benchmarks like MMMU and MMStar.**
>
> A1: Thank you for your concern about benchmark evaluation. We clarify that our assessment focuses on MM-RMs' ability to select highest-quality responses generated by a VLM. Unlike traditional multiple-choice benchmarks, we deliberately selected free-response benchmarks (MM-Vet, Llava-bench, and MMHal-V) for a more appropriate evaluation.
>
> Nevertheless, your suggestion is insightful. We conducted additional evaluations on VL-RewardBench, a benchmark designed to assess MM-RMs.
> |Method|Hallucination|Reason|General|Overall|
> |-|-|-|-|-|
> |MM-RMs (VLFeedback)|||||
> |Standard|56.9|55.7|41.0|54.2|
> |Shortcut-aware|65.0|61.0|44.8|**61.0**|
> |MM-RMs (POVID)|||||
> |Standard|84.6|56.6|63.9|74.5|
> |Shortcut-aware|85.7|57.9|62.3|**75.2**|
> |MM-RMs (RLHF-V)|||||
> |Standard|65.3|57.2|33.3|58.6|
> |Shortcut-aware|75.8|58.8|48.1|**67.4**|
>
> **C2: For the results on Table 2, have the authors tried to combine these three training datasets to train the MM-RM?**
>
> A2: Thank you for this insightful suggestion. We conducted an additional experiment combining equal samples from the three datasets (15K total) for MM-RM training. The evaluation results demonstrated that training with the mixed dataset yielded performance that fell between the individual datasets:
> |Training Set|MM-Vet|Llava-bench|MMHal-V|
> |-|-|-|-|
> |VLFeedback|49.0|80.5|3.70|
> |POVID|46.7|69.6|3.43|
> |RLHF-V|39.9|73.1|3.55|
> |Mixture|47.3|76.5|3.53|
>
> This finding suggests that while combined preference datasets provide diverse training signals, they may partially interfere with each other due to differences in methodologies and preference distributions.
>
> **C3: Can the Shortcut-aware learning algorithm be extended to address other forms of spurious correlations beyond text-only shortcuts? If so, what modifications would be required?**
>
> A3: Thank you for raising an important point about the generalizability of our approach. While our experiments focused on text-only shortcuts due to their prevalence in MM-RMs, the framework's design is intentionally modality-agnostic and can be extended to other forms of spurious correlations with two key modifications:
> 1. Proxy Adaptation: Replace the text-only proxy with a problem-specific proxy (e.g., image-only) to isolate the target spurious feature.
> 2. Dynamic Reweighting: Compute failure signals analogously to Eq. 5 (SFC) for the new proxy, then reweight samples where the proxy struggles. The core mechanism (shifting focus to "hard" multimodal cases) remains unchanged.
>
> We will explicitly discuss this generality in our revisions to highlight the framework's broader applicability.
>
> **C4: It would be helpful to include a comparison with other state-of-the-art methods for improving generalization in multimodal reward models.**
>
> Thank you for your valuable suggestion. We clarify that while text-only RM generalization is well-studied, MM-RMs face a critical research gap despite growing importance. Due to space limitations, we follow ICML’s guidelines and recommend that you refer to our response to Reviewer Rc2L (C5).
>
> **C5: How does the method handle scenarios where multimodal preference datasets are limited or unavailable? Are there any alternative strategies to ensure robust generalization in such cases?**
>
> A5: Thank you for raising this practical consideration. We'd like to clarify that:
> * Preference datasets are fundamentally necessary for RM development, not a limitation specific to our method but rather an inherent requirement of the existing paradigm itself. Any approach to developing MM-RMs will require some form of preference data to learn what constitutes high-quality responses.
> * A key strength of our approach is that it improves bias mitigation without requiring additional data beyond what existing methods already use. Without these innovations, addressing generalization issues would demand significantly more labeled data.
>
> **C6: Could we potentially avoid phenomena like text-only shortcuts by better constructing the Preference dataset? Or could this improve the generalization performance of the MM-RMs trained with it?**
>
> A6: Thank you for highlighting dataset quality as another path to improving generalization. While more balanced datasets can help mitigate shortcuts, achieving truly unbiased data demands extensive human verification to eliminate subtle shortcut patterns—an inherently challenging task.
>
> Our solution addresses this without requiring additional data collection or expensive curation. It works with existing datasets to reduce shortcut learning, offering immediate benefits while minimizing costs. Rather than competing with dataset improvements, our method complements them, strengthening any preference dataset—including those refined in the future.
>
> Please let us know if you have any additional questions.

---

> > ### Comment · Reviewer_Dzcf · 2025-04-06
> >
> > Thank you for the detailed responses. I’ve updated my score to Weak Accept.
> >
> > One more question: Do you plan to open-source your model to benefit the community?

---

> > > ### Author Response · Authors · 2025-04-06
> > >
> > > We sincerely appreciate you for raising your score. As for your question, we are committed to open-sourcing both our model and code to contribute to the research community.

---

### Decision · Program_Chairs · 2025-05-01

**Decision:**

Accept (poster)

**Comment:**

Quoting reviewer Dzcf:

"The paper proposes to improve the generalization of Multimodal Reward Models (MM-RMs) by addressing the issue of unimodal spurious correlations. It introduces a Shortcut-aware MM-RM learning algorithm that dynamically reweights training samples to emphasize multimodal understanding, reducing reliance on text-only shortcuts. Experiments show the effectiveness in generalization, downstream task performance, and scalability."

All reviewers are supportive of accepting the paper.